# Very High Thermotolerance of an Adaptive Evolved *Saccharomyces cerevisiae* in Cellulosic Ethanol Fermentation

**Bin Zhang** [1,†], **Mesfin Geberekidan** [1,†], **Zhao Yan** [1], **Xia Yi** [2] **and Jie Bao** [1,*]

1 State Key Laboratory of Bioreactor Engineering, East China University of Science and Technology, 130 Meilong Road, Shanghai 200237, China
2 National-Local Joint Engineering Research Center for Biomass Refining and High-Quality Utilization, Changzhou University, Changzhou 213164, China
* Correspondence: jbao@ecust.edu.cn
† These authors contributed equally to this work.

**Abstract:** High thermotolerance is an important property of *Saccharomyces cerevisiae* for stable and efficient bioethanol production, especially for large-scale bioethanol production with weak heat transfer and the simultaneous saccharification and fermentation (SSF) of lignocellulosic biomass at high temperatures (above 40 °C). Despite extensive studies involving metabolic engineering and chemical mutagenesis, the improvement of thermotolerance in *S. cerevisiae* under harsh thermal stress (42–45 °C) has been limited. A highly thermotolerant strain, *S. cerevisiae* Z100, by a 91 days' laboratory adaptive evolution in wheat straw hydrolysate was applied for cellulosic ethanol fermentation. The results showed that the cell survival ratio of *S. cerevisiae* Z100 at 50 °C improved by 1.2 times that of the parental strain. The improved thermotolerance of *S. cerevisiae* Z100 at 50 °C was found to contribute significantly to enhanced cellulosic ethanol fermentability. The ethanol production of *S. cerevisiae* Z100 increased by 35%, 127%, and 64% when using wheat straw as feedstock after being maintained at 50 °C for 12 h, 24 h, and 48 h, respectively. Transcriptome analyses suggested that the enhanced trehalose and glycogen synthesis, as well as carbon metabolism, potentially contributed to the improved thermotolerance and the ethanol fermentability of *S. cerevisiae* Z100. This study provides evidence that adaptive evolution is an effective method for increasing the thermotolerance of the *S. cerevisiae* strain for stable and efficient cellulosic ethanol production.

**Keywords:** thermotolerance; adaptive evolution; *Saccharomyces cerevisiae*; cellulosic ethanol

## 1. Introduction

Industrial ethanol fermenters may have a capacity of several thousand cubic meters. In these large fermenters, non-uniform heat transfer may cause localized high-temperature stress and overheating [1]. Thermal stress during cellulosic ethanol fermentation inevitably weakens microbial cell viability, increases the risk of bacterial contamination, and ultimately decreases ethanol fermentation efficiency [2–5]. Therefore, it is crucially important to use a robust strain with high thermotolerance for cellulosic ethanol fermentation in large-scale fermenters.

Extensive studies have been conducted to improve the thermotolerance of *Saccharomyces cerevisiae* through metabolic engineering [6–9] and chemical mutagenesis [10,11]. However, these efforts were often ineffective for improving thermotolerance of *S. cerevisiae* under harsh thermal stress (42–45 °C). On the other hand, adaptive evolution under mild temperatures (35–42 °C) produced certain positive results and improved the thermotolerance of *S. cerevisiae* [1,12–16].

In this work, a highly thermotolerant strain, *S. cerevisiae* Z100, by a 91 days' laboratory adaptive evolution in wheat straw hydrolysate was applied for cellulosic ethanol fermentation. The improved thermotolerance of *S. cerevisiae* Z100 at 50 °C was found to contribute

significantly to the enhanced cellulosic ethanol fermentability. This study provides evidence that adaptive evolution is an effective method for increasing the thermotolerance of *S. cerevisiae* strain for stable and efficient cellulosic ethanol production.

## 2. Materials and Methods

### 2.1. Enzymes and Reagents

Cellic CTec 2.0 enzyme was purchased from Novozymes China (Beijing, China). The protein concentration was 79.9 mg/mL [17]. The other analytical grade chemicals were from Sinopharm Chemical Reagents (Shanghai, China).

### 2.2. Strains and Culture

The parental strain, *S. cerevisiae* XH7, was derived from the wild-type diploid *S. cerevisiae* strain BSIF by integrating the xylose isomerase gene *Ru-xylA* in the genome [18,19]. The adaptively evolved strain, *S. cerevisiae* Z100, was obtained from *S. cerevisiae* XH7 after 91 day's laboratory adaptive evolution in wheat straw hydrolysates. The thermotolerance of this evolved strain, *S. cerevisiae* Z100, is genetically stable. Then, the evolved strain, *S. cerevisiae* Z100, was sent and stored in China General Microbiological Culture Collection Center (CGMCC) with the registration number 17734. The seed culture of *S. cerevisiae* strains was prepared in yeast extract peptone dextrose (YPD) medium and wheat straw hydrolysates. *Amorphotheca resinae* ZN1 (CGMCC No. 7452) was used for biodetoxification [20].

### 2.3. Biorefinery Processing

Wheat straw was harvested from Nanyang, Henan, China, in the fall of 2018. The composition of raw or detoxified wheat straw was determined by the previous protocol [21]. The raw wheat straw contained $32.9 \pm 0.1\%$ (*w/w*) of cellulose and $23.3 \pm 0.2\%$ (*w/w*) of hemicellulose. The pre-handling and dry acid pretreatment of wheat straw was conducted according to previous studies [22]. The harsh acid pretreatment process generated lots of small molecule phenolic aldehydes (e.g., furfural, hydroxymethylfurfural) and weak organic acids (e.g., acetic acid), which strongly inhibit the growth and metabolism of the fermentation strain. Therefore, solid biodetoxification was followed to remove these toxic compounds. The pretreated wheat straw was aerobically biodetoxified with *A. resinae* ZN1 in a 15 L bioreactor until no acetic acid, furfural, and hydroxymethylfurfural were measured (~48 h) [20]. The biodetoxified wheat straw containing 0.36 g cellulose/g dry matter (DM) and 0.12 g xylose/g DM was used for cellulosic ethanol fermentation.

### 2.4. Adaptive Evolution

The pretreated and biodetoxified wheat straw was enzymatically hydrolyzed for 12 h at 150 rpm with 25% (*w/w*) solids content at 50 °C by adding 10 mg cellulase protein/g cellulose. We used 50 mL of wheat straw hydrolysates for adaptive evolution without removing the solid residues, and the initial pH was adjusted to 5.5 with 4 M NaOH. Each transfer was conducted after culturing for 24 h in 50 mL of wheat straw hydrolysate at 35–42 °C, 150 rpm in 250 mL flasks. A total of 5 mL of broth culture containing *S. cerevisiae* cells was pipetted as the seed of the next round of adaptive evolution and inoculated into 50 mL of fresh wheat straw hydrolysate. At the end of adaptive evolution, 100 μL of culture broth was diluted and cultured on agar gel for colony selection and phenotypic testing. The transfer was successfully performed 90 times.

### 2.5. Cellulosic Ethanol Fermentation

There are three sequential steps in cellulosic ethanol fermentation to evaluate the thermotolerance of *S. cerevisiae* strains in 5.0 L fermenter. First, the wheat straw was pre-hydrolyzed into liquid hydrolysate at 50 °C, 150 rpm, 10 mg protein/g cellulose, and 30% (*w/w*) solids loading for 4 h [23]. Then, the seed culture of two *S. cerevisiae* strains (parental strain and evolved strain) with the same cell viability was separately inoculated into the

pre-hydrolyzed wheat straw and cultured for 12 h, 24 h, 48 h at 50 °C, respectively. The *S. cerevisiae* strains were unable to produce ethanol under high temperature stress. Finally, the temperature was naturally cooled to 30 °C, and the ethanol fermentation started. The fermentation pH was maintained at 5.5 by automatic feeding of 4 M NaOH [19].

*2.6. Transcriptome Analyses*

RNA sequencing was performed by CapitalBio Technology Co., Beijing, China. The cDNA library was prepared using NEBNext Ultra RNA Library Prep kit. Pairing and sequencing were performed on an Illumina NovaSeq 6000 platform according to the genome of *S. cerevisiae* S288c (NCBI accession no. GCA_00146045.2).

*2.7. Analysis*

Glucose, xylose, glycerol, ethanol, acetic acid, furfural, and hydroxymethylfurfural were analyzed on HPLC according to the method by Liu et al. [19]. Cell survival of *S. cerevisiae* was assessed by counting the colony-forming units (CFU). Briefly, 100 μL of the $10^{-5}$ or $10^{-6}$ fermentation broth sampling was spread on YPD plates and cultured at 30 °C for 48 h.

**3. Results and Discussions**

*3.1. Thermotolerance of the Adaptively Evolved S. cerevisiae Z100 at 50 °C*

The thermotolerance capacity was assayed by maintaining parental *S. cerevisiae* XH7 and evolved *S. cerevisiae* Z100 at 50 °C for 12 h, 24 h, or 48 h, respectively (Figure 1). Compared with the parental strain, the cell survival of *S. cerevisiae* Z100 was significantly increased (Figure 1a), suggesting that laboratory adaptive evolution brought about the improved thermotolerance of the parental strain. The glycerol generation of *S. cerevisiae* Z100 at 50 °C was increased by 1.5- and 3.1-fold that of the parental strain (Figure 1b). Glycerol plays a pivotal role in maintaining redox homeostasis and alleviates the redox unbalance under high-temperature stress in *S. cerevisiae* [24–27]. The improved cell survival and glycerol generation at 50 °C suggested the higher thermotolerance of *S. cerevisiae* Z100 to the parental strain.

*3.2. Improved Thermotolerance of S. cerevisiae Z100 Facilitated Cellulosic Ethanol Fermentation*

Ethanol fermentation of the adaptively evolved *S. cerevisiae* Z100 was evaluated after being maintained at 50 °C for 12–48 h using pretreated and biodetoxified wheat straw feedstocks under 30% (*w/w*) solids loading (Figure 2). The adaptively evolved *S. cerevisiae* Z100 demonstrated at least 24 h shorter lag phase time of glucose consumption and ethanol generation than the parental strain after being maintained at 50°C for over 12 h. The shortened lag phase time was mainly due to the improved cell survival of the evolved *S. cerevisiae* Z100 after thermal stress. As for xylose consumption, *S. cerevisiae* Z100 utilized at least 77% xylose after being maintained at 50 °C for 12–48 h, whereas xylose utilization of the parental strain was less than 33%. *S. cerevisiae* had a priority to utilize glucose rather than xylose [18], and the shortened lag phase time of glucose consumption benefited the metabolism of xylose in the evolved *S. cerevisiae* Z100. The higher xylose consumption in evolved *S. cerevisiae* Z100 contributed to the improvement of ethanol production. The ethanol production of *S. cerevisiae* Z100, after being maintained at 50 °C, was significantly improved by 35% for 12 h, 130% for 24 h, and 64% for 48 h, respectively, compared to that of the parental strain (Figure 2b). The adaptively evolved *S. cerevisiae* Z100 exhibited significantly improved cell survival at 50 °C and enhanced thermotolerance, which would benefit traditional bioethanol production in large-scale fermenters and SSF at high temperatures.

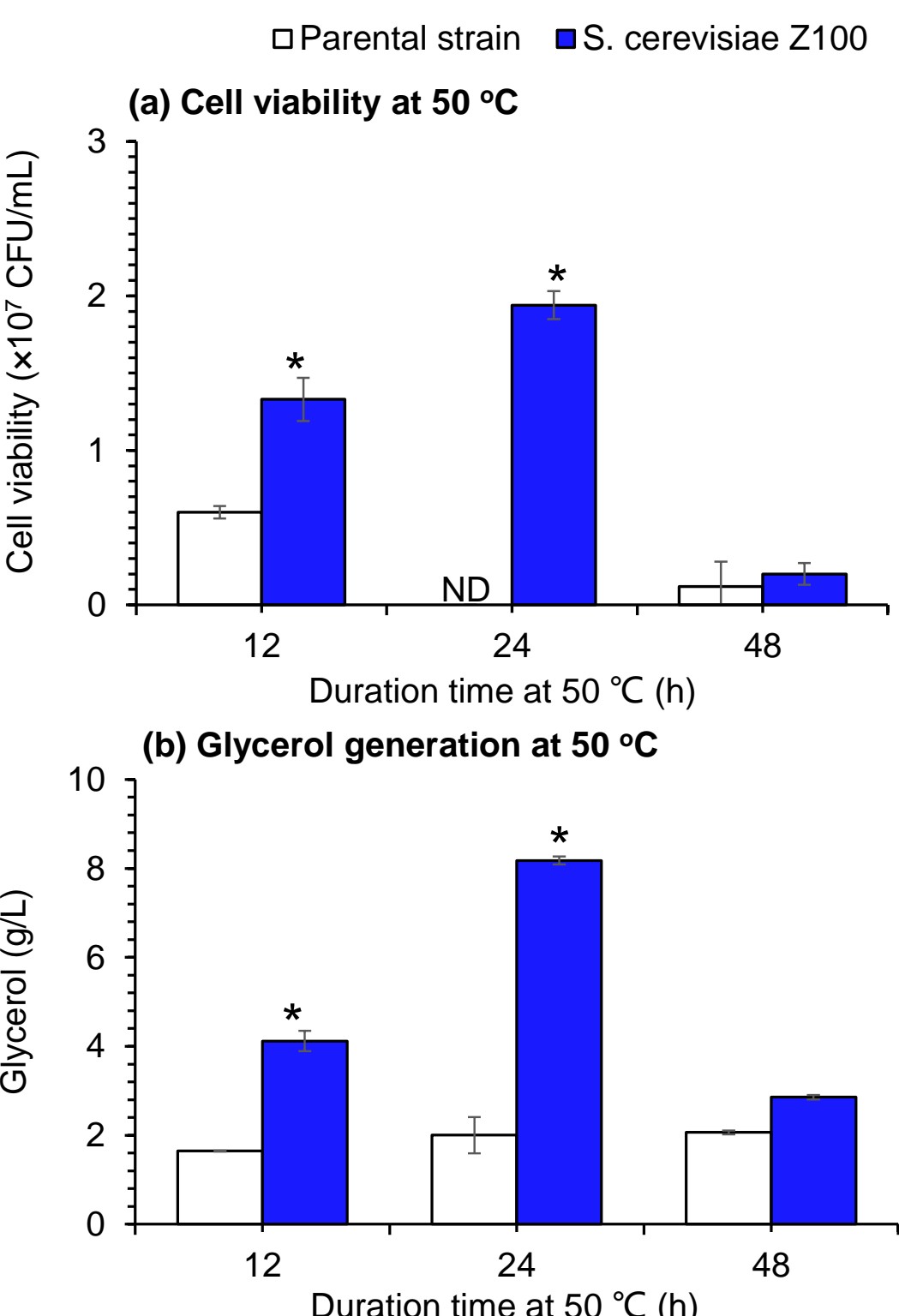

**Figure 1.** Cell survival and glycerol generation of *S. cerevisiae* Z100 at 50 °C. (**a**) Cell survival; (**b**) Glycerol generation. The parental strain, *S. cerevisiae*, XH7 was used as the control. ND (not detected): below the prescribed minimum of detection. The asterisk (*) represents a value significantly different from the control value ($p < 0.05$). Each experiment was performed in triplicate. The error bar represents the standard deviation.

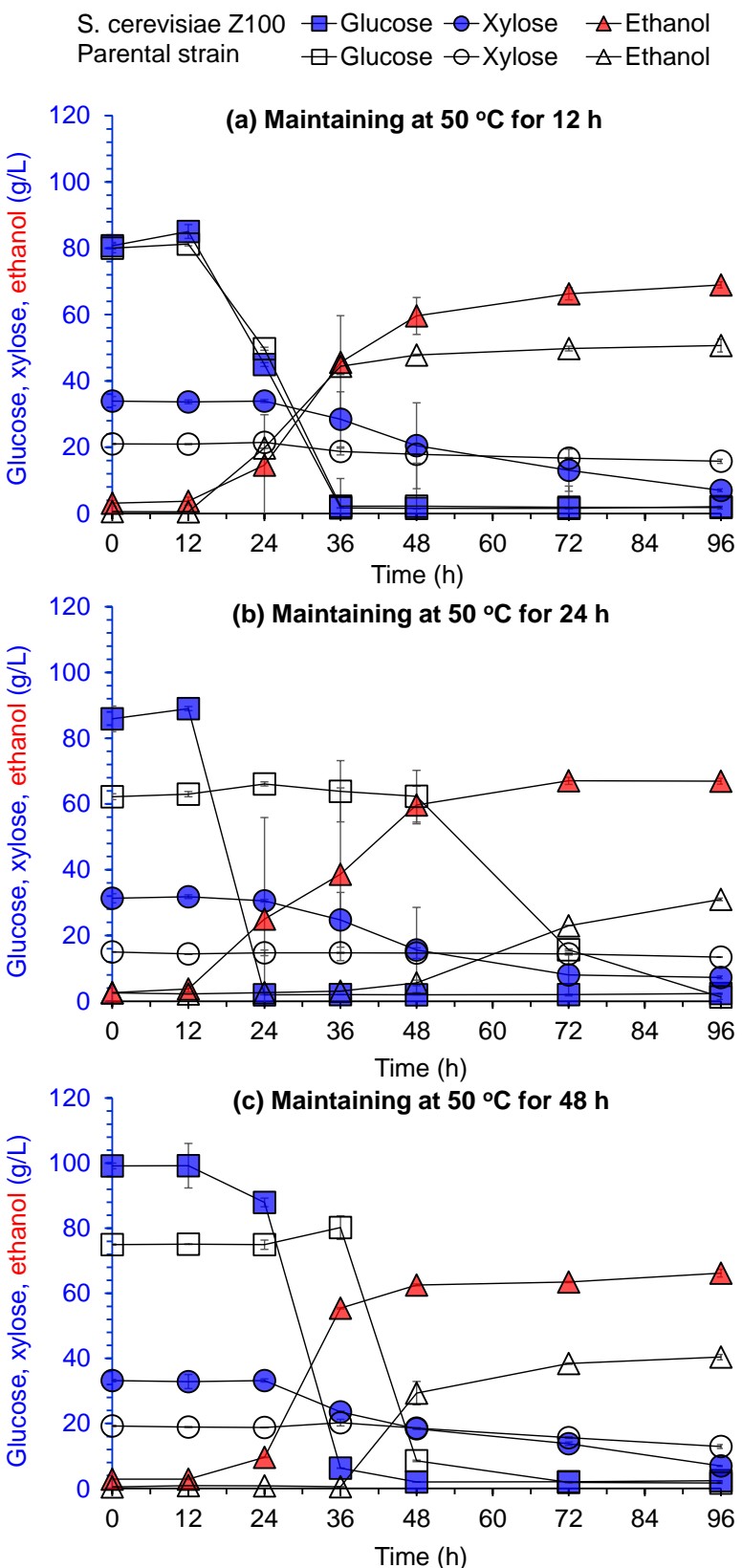

**Figure 2.** Ethanol fermentation of *S. cerevisiae* Z100 after being maintained at 50 °C for different duration times. (**a**) 12 h; (**b**) 24 h; (**c**) 48 h. Ethanol fermentation was conducted at 30 °C and 150 rpm under 30% (*w/w*) wheat straw solids loading. Each experiment was performed in triplicate. The error bar represents the standard deviation.

*3.3. Transcriptome Analyses Revealed the Potential Genes Responsible for Improved Thermotolerance of S. cerevisiae Z100*

In the transcriptional responses to high temperature (30 °C vs. 50 °C) between the evolved strain *S. cerevisiae* Z100 and the parental strain *S. cerevisiae* XH7, the same 17 significantly up-regulated genes and same 10 significantly down-regulated genes were identified in all 4 sets of transcriptional analyses (Figure 3). These differential expression genes, such as heat shock protein (*YFL014W*), NADPH dehydrogenase (*YPL171C*), cytosolic catalase (*YGR088W*), and transcription factors (*YOR028C*, *YIL101C*, *YLR223C*), are potential targets to improve the thermotolerance of *S. cerevisiae* strain by metabolic engineering.

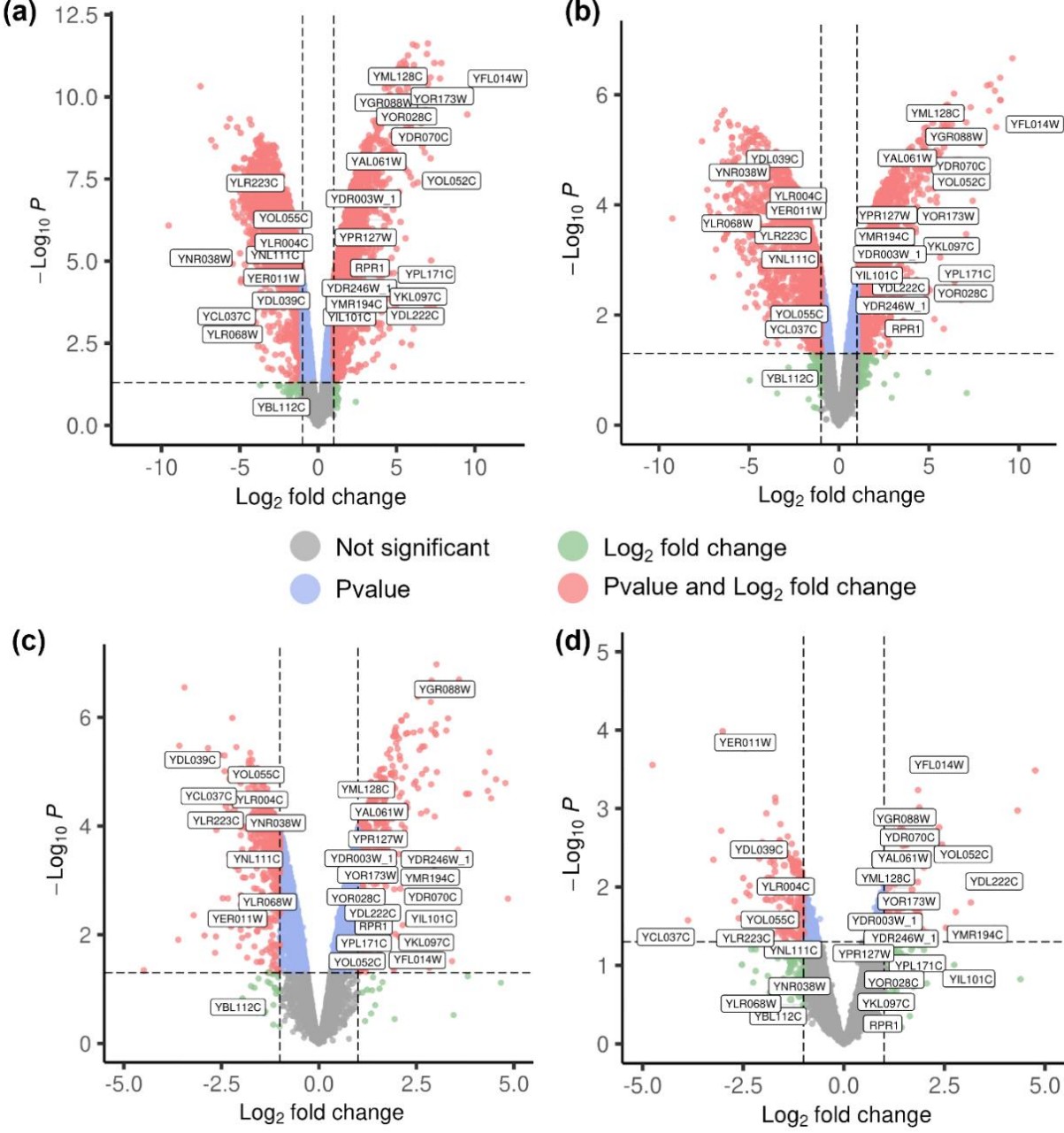

**Figure 3.** Transcriptome analyses at 30 °C and 50 °C. *S. cerevisiae* Z100 at 50 °C compared to 30 °C (**a**)—total: 5971 variables; *S. cerevisiae* XH7 at 50 °C compared to 30 °C (**b**)—total: 5973 variables; *S. cerevisiae* Z100 at 30 °C compared to *S. cerevisiae* XH7 at 30 °C (**c**)—total: 5964 variables; *S. cerevisiae* Z100 at 50 °C compared to *S. cerevisiae* XH7 at 50 °C (**d**)—total: 5955 variables; *S. cerevisiae* Z100 at 30 °C compared to *S. cerevisiae* XH7 at 30 °C.

The gene expression profiling of carbon metabolism In two *S. cerevisiae* strains at 30 °C and 50 °C were further compared (Figure 4). Although more glycerol was generated by *S. cerevisiae* Z100 at a high temperature, the expressions of GPD (glycerol-3-phosphorylase)- and GPP (glycerol-3-phosphate phosphatase)-encoding glycerol synthesis showed unremarkable changes. One possible reason is that the changes in transcriptional level lag behind the direct response and regulation of temperature sensors [28]. The key genes in trehalose and glycogen synthesis, including TSL (trehalose 6-phosphate synthase), TPS (α, α-trehalose-phosphate synthase), GSY2 (glycogen synthase 2), GLC3 (1,4-α-glucan branching enzyme), and PGM2 (phosphoglucomutase), were up-regulated in *S. cerevisiae* Z100, both at 30 °C or 50 °C, compared to the parental strain, *S. cerevisiae* XH7. Trehalose, a protein stabilizer and depressor of non-specific protein aggregation, played an important role against high temperature stress [3,29]. Glycogen, the main reserve sugar, was also responding to high temperature stress by rapidly converting to trehalose [30]. The accumulation of trehalose and glycogen in *S. cerevisiae* Z100 would be effective in response to the upcoming high temperature stress.

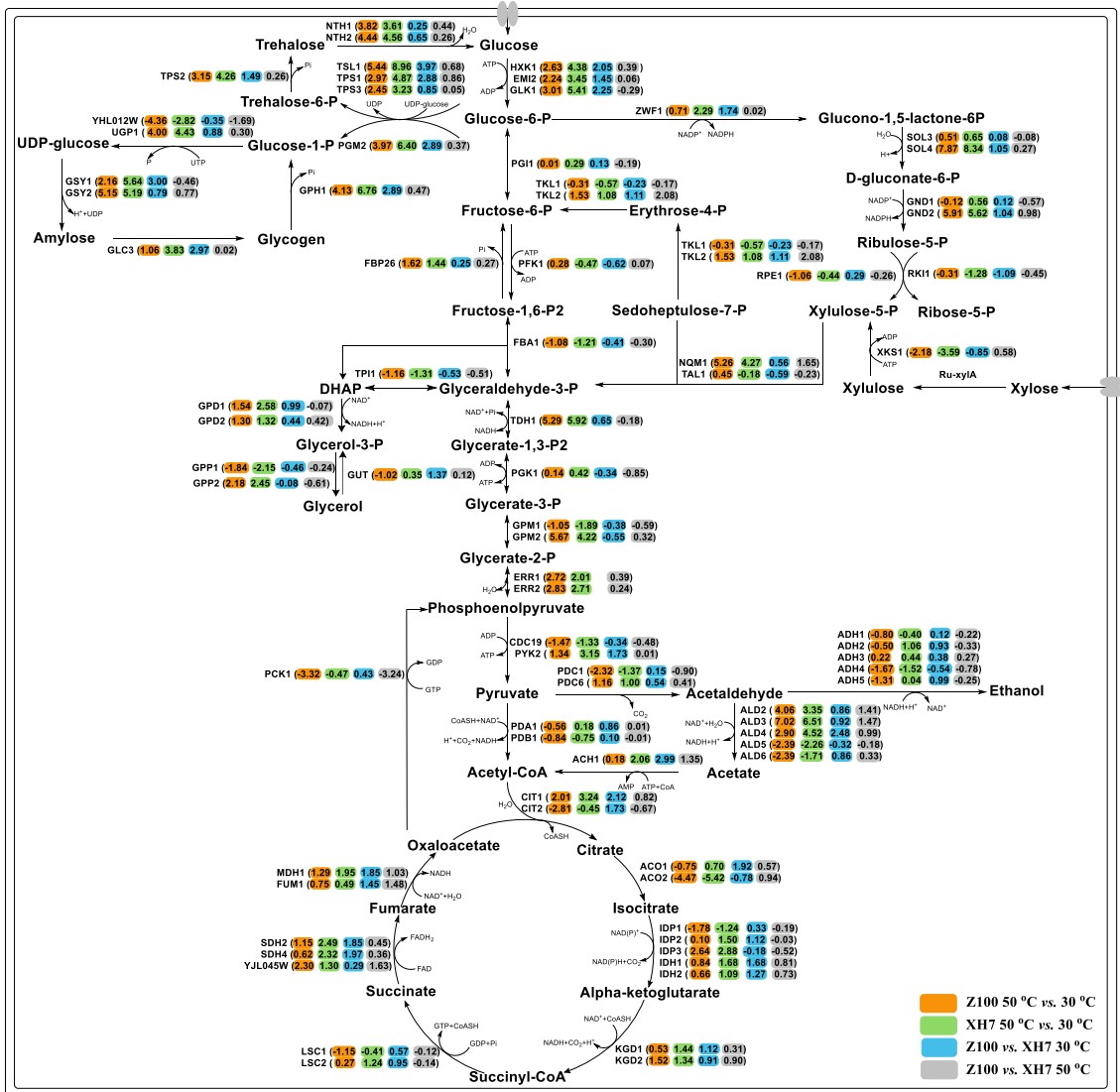

**Figure 4.** Transcriptional landscapes of carbohydrate metabolism for two *S. cerevisiae* strains. The numbers in colored boxes represent the fold change of gene expression for *S. cerevisiae* strains. The positive and negative numbers separately indicate up- and down-regulation, with absolute value of the $\log_2$ ratio $\geq 2.0$.

The key genes in the central carbon metabolism, including GLK1 (glucokinase), HXK1 (hexokinase), PYK2 (pyruvate kinase) involved in the Embden–Meyerhof–Parnas pathway (EMP), CIT1 (citrate synthase) involved in the tricarboxylic acid (TCA) cycle, and ZWF1 (glucose-6-phosphate dehydrogenase) involved in the pentose phosphate pathway (PPP), were also up-regulated in *S. cerevisiae* Z100 at 30 °C. The enhanced carbon metabolism contributed to the better ethanol fermentability of *S. cerevisiae* Z100 compared to the parental strain *S. cerevisiae* XH7.

## 4. Conclusions

*S. cerevisiae* Z100, obtained by long-term laboratory adaptive evolution, showed improved thermotolerance, with 1.2-fold increased cell survival and 1.5-fold more glycerol generation at 50 °C for 12 h. The outstanding thermotolerance in the evolved *S. cerevisiae* Z100 also contributed to the improvement of cellulosic ethanol fermentability, with at least 35% more ethanol production. Transcriptome analyses indicated that the enhanced trehalose and glycogen synthesis, as well as the carbon metabolism, potentially contributed to the improved thermotolerance and ethanol fermentability for the evolved strain *S. cerevisiae* Z100. This study not only provided a robust, thermotolerant *S. cerevisiae*, suitable for large-scale cellulosic ethanol production, but also conducted a practical approach to improving the thermotolerance of ethanologenic strains.

**Author Contributions:** J.B.: funding acquisition, conceptualization, supervision, writing—review and editing; B.Z.: data curation, formal analysis, investigation, writing—original draft; M.G.: data curation, formal analysis, investigation; Z.Y. and X.Y.: conceptualization, writing—original draft. All authors have read and agreed to the published version of the manuscript.

**Funding:** This research was supported by the National Natural Science Foundation of China (21978083, 31961133006), and the Yangfan Project of Science and Technology Committee of Shanghai Municipality (23YF1409900).

**Institutional Review Board Statement:** Not applicable.

**Informed Consent Statement:** Not applicable.

**Data Availability Statement:** Not applicable.

**Conflicts of Interest:** The authors declare no conflict of interest.

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
