# Peer review of "Very High Thermotolerance of an Adaptive Evolved Saccharomyces cerevisiae in Cellulosic Ethanol Fermentation"

_fermentation, doi:10.3390/fermentation9040393_

Round 1

Reviewer 1 Report

This work reports the development of a thermotolerant strain of Saccharomyces cerevisiae by adaptative evolution. The transcriptomic analysis reveals the genes potentially involved in the adaptation. Also, the authors assessed ethanol production from hydrolyzed wheat straw at 50°C. The manuscript is, in general, well-structured and written. Material and methods are clearly written. The results and discussion support the conclusion. However, the authors must address some issues for a publication: 

  • Figures 1 and 2, please indicate what the bars deviations represent. 
  • Improve the quality of Figure 4. Most of the labels are very small and not readable.
  • In conclusion (line 178), correct “adoptive evolution”, must be “adaptative evolution”.

Author Response

Question 1: Figures 1 and 2, please indicate what the bars deviations represent.

Answer: As your suggestion, the explanation of error bar had been added in the legends of Figures 1 and 2 as follows:

“Each experiment was performed in triplicate. The error bar represents the standard deviation.”

Question 2: Improve the quality of Figure 4. Most of the labels are very small and not readable.

Answer: The quality of Figure 4 had been improved as your suggestion.

Question 3: In conclusion (line 178), correct “adoptive evolution”, must be “adaptative evolution”.

Answer: The incorrect spelling had been corrected.

In addition, the manuscript had been carefully reedited. All the changed did not list here but highlight in red in the revised manuscript.

Reviewer 2 Report

Nice and interesting work.

However, some improvements will make the manuscript easier to understand by a larger range of readers.

- On the top of Figure 1 and Figure 2, the label “Parental strain” (not Parenal)

- Line 56 - The origin of the parental strain S. cerevisiae XH7 should be indicated.

- Line 57 - The strain S. cerevisiae Z100 is already registered at the CGMCC - is it already commercially available? The thermotolerance is kept in successive generations of yeasts? The authors used this modified strain or have they developed it by themselves?

- Line 61 – please, give an insight about the concept and the intention of the biodetoxification.

- Line 68 – please, mention what are the main inhibitors that must be removed.

- Lines 71 – 75 (item 2.4) – please, explain better what happened during the 91 day’s laboratory adaptive evolution; did the successive transfers involved cells centrifugation and addition of fresh wheat straw hydrolysate? (91 times?). I wonder if the authors have developed the strain by themselves or if they got the strain from the CGMCC. If this is the case the mention to the “91 days’ laboratory adaptive evolution” should not be referred in the Abstract.

- Figure 2 and item 2.5 – The fermentation includes 3 sequential steps in 5.0 L fermenter at 150 rpm:

            - 1st – 4 h of pre-saccharification at 50ºC with enzyme and without cells;

            - 2nd - 12, 24 or 48 h at 50ºC with cells;

            - 3rd – 96 h of fermentation at 30ºC (profiles in the graphs of the Figure 2);

Is this right? It should be better explained.

Both strains were submitted to the same protocol, right?

- Figure 2 – the symbols are too large and they hide the standard deviation bars in many of them. The meaning of the bars is not explained – it seems that the 3 fermentations were not repeated; so, the standard deviation bars corresponds to different analysis of the compounds by HPLC ?

- Figure 2 – the ethanol production was increased when compared to the use of the parental strain, most probably because the xylose was also consumed; anyway, the thermotolerance increase is always beneficial.

- I am not the right person to evaluate the item 3.3 concerning transcriptome analysis, because it is not my expertise.

Author Response

Question 1: On the top of Figure 1 and Figure 2, the label “Parental strain” (not Parenal)

Answer: The label of “Parental” in Figures 1 and 2 had been corrected to “Parental strain”.

Question 2: The origin of the parental strain S. cerevisiae XH7 should be indicated.

Answer: The origin of the parental strain S. cerevisiae XH7 had been added in Page 4, Lines 69-70 as follows:

“The parental strain S. cerevisiae XH7 was derived from the wild-type diploid S. cerevisiae strain BSIF by integrating the xylose isomerase gene Ru-xylA in the genome [18,19].”

References:

  1. Li, H.X.; Shen, Y.; Wu, M.L.; Hou, J.; Jiao, C.L.; Li, Z.L.; Liu, X.L.; Bao, X.M. Engineering a wild-type diploid Saccharomyces cerevisiae strain for second-generation bioethanol production. Bioresour. Bioprocess. 2016, 3, 51. https://doi.org/10.1186/s40643-016-0126-4.
  2. Liu, G.; Zhang, Q.; Li, H.X.; Qureshi, A.S.; Zhang, J.; Bao, X.M.; Bao, J. Dry biorefining maximizes the potentials of simultaneous saccharification and co-fermentation for cellulosic ethanol production. Biotechnol. Bioeng. 2018, 115, 60-69. https://doi.org/10.1002/bit.26444.

Question 3: Line 57 - The strain S. cerevisiae Z100 is already registered at the CGMCC - is it already commercially available? The thermotolerance is kept in successive generations of yeasts? The authors used this modified strain or have they developed it by themselves?

Answer: The strain S. cerevisiae Z100 was obtained by long-term laboratory adaptive evolution carried out by ourselves. Then the strain S. cerevisiae Z100 was sent and stored in CGMCC with the registration number of 17734, but not commercially available.

The thermotolerance of this evolved strain S. cerevisiae Z100 is genetically stable, because the whole-genome re-sequencing showed that S. cerevisiae Z100 strain actually have lots of mutations in genome compared to the parental strain. The detailed description had been added in Page 4, Lines 72-75 as follows:

“The thermotolerance of this evolved strain S. cerevisiae Z100 is genetically stable. Then the evolved strain S. cerevisiae Z100 was sent and stored in China General Microbiological Culture Collection Center (CGMCC) with the registration number 17734.”

Question 4: Line 61 – please, give an insight about the concept and the intention of the biodetoxification.

Answer: Toxic compounds are generated during the harsh acid pretreatment of wheat straw, including furan aldehydes from excessive pentose and hexose degradation, weak organic acids from acetyl groups oxidation, or phenolic aldehydes from lignin degradation. These inhibitory compounds strongly inhibit the viability and metabolism of microorganisms used in subsequent fermentation step. Therefore, a detoxification or conditioning step must be followed to remove the inhibitors. Biological detoxification (biodetoxification) using specific microorganisms is considered as the most promising approach for detoxification among the various available options by its great effectiveness for completely and ultimately degrading inhibitors into CO2 and water while preserving fermentable sugars. The concept and intention of biodetoxification had been added in Page 4, Lines 84-87 as follows:

“The harsh acid pretreatment process generated lots of small molecule phenolic aldehydes (e.g., furfural, hydroxymethylfurfural) and weak organic acids (e.g., acetic acid), which strongly inhibit the growth and metabolism of the fermentation strain. Therefore, the solid biodetoxification was followed to remove these toxic compounds.”

Question 5: - Line 68 – please, mention what are the main inhibitors that must be removed.

Answer: The biodetoxification was lasted until no acetic acid, furfural, and hydroxymethylfurfural was measured (~48 h). The description had been added in Page 5, Line 89 as follows:

“The pretreated wheat straw was aerobically biodetoxified with A. resinae ZN1 in a 15 L bioreactor until no acetic acid, furfural, and hydroxymethylfurfural was measured (~48 h).”

Question 6: - Lines 71 – 75 (item 2.4) – please, explain better what happened during the 91 day’s laboratory adaptive evolution; did the successive transfers involved cells centrifugation and addition of fresh wheat straw hydrolysate? (91 times?). I wonder if the authors have developed the strain by themselves or if they got the strain from the CGMCC. If this is the case the mention to the “91 days’ laboratory adaptive evolution” should not be referred in the Abstract.

Answer: The strain S. cerevisiae Z100 was obtained through adaptive evolution carried out by ourselves, rather than directly purchased. Each transfer was conducted after culturing for 24 h in wheat straw hydrolysate. 5 mL of broth culture containing S. cerevisiae Z100 cells were pipetted as the seed, and inoculated into 50 mL of fresh wheat straw. The detailed description about the adaptive evolution had been added in Page 5, Lines 96-102 as follows:

“Each transfer was conducted after culturing for 24 h in 50 mL of wheat straw hydrolysate at 35-42 ℃, 150 rpm in 250 mL flasks. 5 mL of broth culture containing S. cerevisiae cells were pipetted as the seed of next round of adaptive evolution, and inoculated into 50 mL of fresh wheat straw hydrolysate. At the end of adaptive evolution, 100 μL of culture broth was diluted and cultured on agar gel for colony selection and phenotypic testing. The transfer was successfully performed 90 times.”

Question 7: - Figure 2 and item 2.5 – The fermentation includes 3 sequential steps in 5.0 L fermenter at 150 rpm:

         - 1st – 4 h of pre-saccharification at 50ºC with enzyme and without cells;

         - 2nd - 12, 24 or 48 h at 50ºC with cells;

         - 3rd – 96 h of fermentation at 30ºC (profiles in the graphs of the Figure 2);

Is this right? It should be better explained.

Both strains were submitted to the same protocol, right?

Answer: As you mentioned, there are three sequential steps in cellulosic ethanol fermentation to evaluate the thermotolerance of S. cerevisiae strains in 5.0 L fermentor. First, the wheat straw was pre-hydrolyzed into liquid hydrolysate at 50 ºC, 150 rpm, 10 mg protein/g cellulose, 30% (w/w) solids loading for 4 h. Then the seed culture of two S. cerevisiae strains (parental strain and evolved strain) with the same cell viability was separately inoculated into the pre-hydrolyzed wheat straw, and cultured for 12 h, 24 h, 48 h, respectively. The S. cerevisiae strains were unable to produce ethanol under high temperature stress. Finally, the temperature was naturally cooled to 30 ℃, and the ethanol fermentation started. The fermentation pH was maintained at 5.5 by automatic feeding of 4 M NaOH. Both strains were submitted to the same protocol. The detailed description had been added in Page 5, Lines 104-112 as follows:

“There are three sequential steps in cellulosic ethanol fermentation to evaluate the thermotolerance of S. cerevisiae strains in 5.0 L fermenter. First, the wheat straw was pre-hydrolyzed into liquid hydrolysate at 50 ºC, 150 rpm, 10 mg protein/g cellulose, 30% (w/w) solids loading for 4 h [23]. Then the seed culture of two S. cerevisiae strains (parental strain and evolved strain) with the same cell viability was separately inoculated into the pre-hydrolyzed wheat straw, and cultured for 12 h, 24 h, 48 h at 50 ºC, respectively. The S. cerevisiae strains were unable to produce ethanol under high temperature stress. Finally, the temperature was naturally cooled to 30 ℃, and the ethanol fermentation started. The fermentation pH was maintained at 5.5 by automatic feeding of 4 M NaOH [19].”

References:

  1. Liu, G.; Zhang, Q.; Li, H.X.; Qureshi, A.S.; Zhang, J.; Bao, X.M.; Bao, J. Dry biorefining maximizes the potentials of simultaneous saccharification and co-fermentation for cellulosic ethanol production. Biotechnol. Bioeng. 2018, 115, 60-69. https://doi.org/10.1002/bit.26444.
  2. Zhang., B.; Khushik, F.A.; Zhan, B.; Bao, J. Transformation of lignocellulose to starch-like carbohydrates by organic acid-catalyzed pretreatment and biological detoxification. Biotechnol. Bioeng. 2021, 118, 4105-4118.

Question 8: - Figure 2 – the symbols are too large and they hide the standard deviation bars in many of them. The meaning of the bars is not explained – it seems that the 3 fermentations were not repeated; so, the standard deviation bars corresponds to different analysis of the compounds by HPLC?

Answer: Figure 2 had been revised according to your suggestion. The explanation of error bar had been added in the legends of Figures 1 and 2 as follows:

“Each experiment was performed in triplicate. The error bar represents the standard deviation.”

Question 9: - Figure 2 – the ethanol production was increased when compared to the use of the parental strain, most probably because the xylose was also consumed; anyway, the thermotolerance increase is always beneficial.

Answer: As your suggestion, we had added the corresponding explanation in Page 8, Lines 155-156 as follows:

“The higher xylose consumption in evolved S. cerevisiae Z100 contributed to the improvement of ethanol production.”

In addition, the manuscript had been carefully reedited. All the changed did not list here but highlight in red in the revised manuscript.

Round 2

Reviewer 2 Report

The manuscript has been quite improved and is now much clearer to understand. In my opinion it deserves to be published in Fermentation journal.